# Quantum speedup in the identification of cause–effect relations

Giulio Chiribella[1,2,3] & Daniel Ebler[1,4]

The ability to identify cause–effect relations is an essential component of the scientific method. The identification of causal relations is generally accomplished through statistical trials where alternative hypotheses are tested against each other. Traditionally, such trials have been based on classical statistics. However, classical statistics becomes inadequate at the quantum scale, where a richer spectrum of causal relations is accessible. Here we show that quantum strategies can greatly speed up the identification of causal relations. We analyse the task of identifying the effect of a given variable, and we show that the optimal quantum strategy beats all classical strategies by running multiple equivalent tests in a quantum superposition. The same working principle leads to advantages in the detection of a causal link between two variables, and in the identification of the cause of a given variable.

[1] Department of Computer Science, The University of Hong Kong, Pokfulam Road, Hong Kong. [2] Department of Computer Science, University of Oxford, Oxford OX1 3QD, UK. [3] Perimeter Institute for Theoretical Physics, Waterloo, ON N2L 2Y5, Canada. [4] Department of Physics, Institute for Quantum Science and Engineering, Southern University of Science and Technology, Shenzhen 518055, China. Correspondence and requests for materials should be addressed to G.C. (email: giulio@cs.hku.hk)

Identifying causal relations is a fundamental primitive in a variety of areas, including machine learning, medicine, and genetics[1–3]. A canonical approach is to formulate different hypotheses on the cause–effect relations characterizing a given phenomenon, and test them against each other. For example, in a drug test some patients are administered the drug, while others are administered a placebo, with the scope of determining whether or not the drug causes recovery. Traditionally, causal discovery techniques have been based on classical statistics, which effectively describes the behavior of macroscopic variables. However, classical techniques become inadequate when dealing with quantum systems, whose response to interventions can strikingly differ from that of classical random variables[4,5].

Recently, there has been a growing interest in the extension of causal reasoning to the quantum domain. Several quantum generalizations of the notion of causal network have been proposed[6–15] and new algorithms for quantum causal discovery have been designed[16–20]. Besides its foundational relevance, the study of quantum causal discovery algorithms is expected to have applications in the emerging area of quantum machine learning[21,22], in the same way as classical causal discovery algorithms have previously impacted classical artificial intelligence.

An intriguing possibility is that quantum mechanics may provide enhanced ways to identify causal links. A clue in this direction comes from refs. [17,18], where the authors show that certain quantum correlations are witnesses of causal relationships, in apparent violation of the classical tenet "correlation does not imply causation". This observation suggests that quantum setups for testing causal relationships could overcome some of the limitations of existing classical setups. However, the type of advantage highlighted in refs. [17,18] only concerns a limited class of setups, where the experimenter is constrained to a subset of the possible interventions. If arbitrary interventions are allowed, this particular type of advantage disappears. A fundamental open question is whether quantum setups can offer an advantage over all classical setups, without any restriction on the experimenter's interventions.

Here, we answer the question in the affirmative, proving that quantum features like superposition and entanglement can significantly speed up the identification of causal relations. We start from the task of deciding which variable, out of a list of candidates, is the effect of a given variable. We first analyze the problem in the classical setting, determining the performance of the best classical strategy. Then, we construct a quantum strategy that reduces the error probability by an exponential amount, doubling the decay rate of the error probability with the number of accesses to the relevant variables. Remarkably, the decay rate of our strategy is the highest achievable rate allowed by quantum mechanics, even if one allows for exotic setups where the order of operations is indefinite[23,24]. The key ingredient of the quantum speedup is the ability to run multiple equivalent experiments in a quantum superposition. The same working principle enables quantum speedups in a broader set of tasks, including, e.g., the task of deciding whether there exists a causal link between two given variables, and the task of identifying the cause of a given variable.

## Results

### Theory-independent framework for testing causal hypotheses.
Here, we outline a framework for testing causal hypotheses in general physical theories[25–30]. In this framework, variables are represented as physical systems, each system with its set of states. The framework applies to theories satisfying the Causality Axiom[28], stating that the probability of an event at a given time should not depend on choices of settings made at future times.

A causal relation between variable $A$ and variable $B$ is represented by a map describing how the state of $B$ responds to changes in the state of $A$. If the map discards $A$ and outputs a fixed state of $B$, then no causal influence can be observed. In all the other cases, some change of $A$ will lead to an observable change of $B$. Hence, we say that $A$ is a cause for $B$.

In general, the set of allowed causal relationships depends on the physical theory, which determines which maps can be implemented by physical processes. In classical physics, cause–effect relations can be represented by conditional probability distributions of the form $p(b|a)$, where $a$ and $b$ are the values of the random variables $A$ and $B$, respectively. In quantum theory, cause–effect relations are described by quantum channels, i.e., completely positive trace-preserving maps transforming density matrices of system $A$ into density matrices of system $B$.

Given a set of variables, one can formulate hypotheses on the causal relationships among them. For example, consider a three-variable scenario, where variable $A$ may cause either variable $B$ or variable $C$, but not both. The causal relation is described by a process $\mathcal{C}$, with input $A$ and outputs $B$ and $C$. Here, we consider two alternative causal hypotheses: either $A$ causes $B$ but not $C$; or $A$ causes $C$ but not $B$. The problem is to distinguish between these two hypotheses without having further knowledge of the physical process responsible for the causal relation. This means that the process $\mathcal{C}$ is unknown, except for the fact that it must compatible with one and only one of the two hypotheses. Mathematically, the two hypotheses correspond to two sets of physical processes, and the problem is to determine which set contains the process $\mathcal{C}$.

In order to decide which hypothesis is correct, we assume that the experimenter has black box access to the physical process $\mathcal{C}$. The experimenter can probe the process for $N$ times, intervening between one instance and the next, as illustrated in Fig. 1. In the end, a measurement is performed and its outcome is used to guess the correct hypothesis.

An important question is how fast the probability of error decays with $N$. The decay is typically exponential, with an error probability vanishing as $p_{\mathrm{err}}(N) \approx 2^{-RN}$ for some positive constant $R$, which we call the discrimination rate. The operational meaning of the discrimination rate is the following. Given an error threshold $\varepsilon$, the error probability can be made smaller than $\varepsilon$ using approximately $N > \log \varepsilon^{-1}/R$ calls to the unknown process. The bigger the rate, the smaller the number of calls needed to bring the error below the desired threshold.

Since the explicit form of the process $\mathcal{C}$ is unknown, we take $p_{\mathrm{err}}(N)$ to be the worst-case probability over all processes compatible with the two given causal hypotheses. If prior information over $\mathcal{C}$ is available, one may also consider a weaker performance measure, based on the average with respect to some

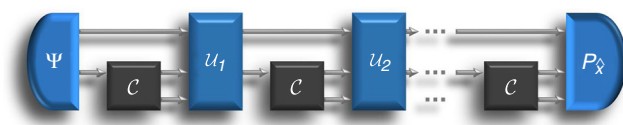

**Fig. 1** Testing causal hypotheses in the black box scenario. The unknown process $\mathcal{C}$ induces a causal relation between one input variable and two output variables. The experimenter probes the process for $N$ times, intervening on the relevant variables at each time step. The first intervention is the preparation of a state $\Psi$, involving the input of the black box and, possibly, an additional reference system (top wire). The subsequent interventions $\mathcal{U}_i$ manipulate the output variables and prepare the inputs variables for the next steps. In the end, the output variables and the reference system are measured, and the measurement outcome is used to infer the causal relation

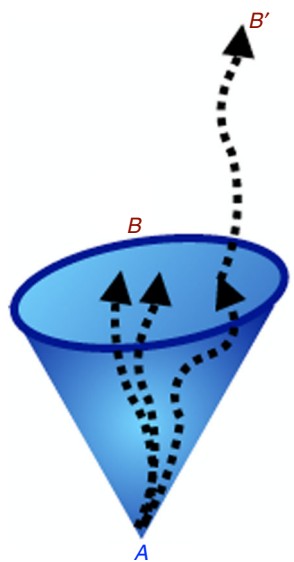

**Fig. 2** Spacetime picture of a causal intermediary. Variable $A$ is localized at a point in spacetime, and its causal influences propagate within its future light cone. Variable $B$ is distributed over a section of the light cone of $A$ and intercepts all the influences of $A$. Every other variable $B'$ that is affected by $A$ and comes after $B$ must be obtained from variable $B$ through some physical process

prior. In the following, we stick to the worst case scenario, as it provides a stronger guarantee on the performance of the test.

**Identifying causal intermediaries**. A variable $B$ is a causal intermediary for variable $A$ if all the influences of $A$ propagate through $B$. Physically, one can think of $B$ as a slice of the future light cone of $A$, so that all causal influences of $A$ must pass through $B$, as illustrated in Fig. 2. Mathematically, the fact that $B$ is a causal intermediary means that there exists a process $\mathcal{C}$ from $A$ to $B$ such that for every other variable $B'$ and for every process $\mathcal{C}'$ with input $A$ and output $B'$ one can decompose $\mathcal{C}'$ as $\mathcal{C}' = \mathcal{R} \circ \mathcal{C}$, where $\mathcal{R}$ is a suitable process from $B$ to $B'$.

The condition that a variable is a causal intermediary of another has a simple characterization in all physical theories where processes are fundamentally reversible, meaning that they can be modeled as the result of a reversible evolution of the system and an environment[28]. The reversibility condition is captured by the expression $\mathcal{C} = (\mathcal{I}_B \otimes \mathrm{Tr}_{E'})\mathcal{U}(\mathcal{I}_A \otimes \eta_E)$, where variables $E$ and $E'$ represent the environment (before and after the interaction), $\eta$ is the initial state of the environment, $Tr_{E'}$ is the operation of discarding system $E'$[28], and $\mathcal{U}$ is a reversible process from $AE$ to $BE'$.

When the reversibility condition is satisfied, the variable $A$ can be recovered from variables $B$ and $E'$. If variable $B$ is to be a causal intermediary of $A$, then the process $\mathcal{C}$ must be correctable, in the sense that its action can be undone by another process $\mathcal{R}$. In addition, if the state spaces of variables $A$ and $B$ are finite dimensional and of the same dimension, then the process $\mathcal{C}$ must be reversible. In classical theory, this means that $\mathcal{C}$ is an invertible function. In quantum theory, this means that $\mathcal{C}$ is a unitary channel, of the form $\mathcal{C}(\rho) = U\rho U^\dagger$ for some unitary operator $U$.

In the following, we will consider the task of identifying which variable, out of a given set of candidates, is the causal intermediary of a given variable $A$. An important feature of this task is that it admits a complete analytical treatment, allowing us to rigorously prove a quantum advantage over all classical strategies. Besides its fundamental interest, this advantage could

have applications to the task of monitoring the information flow in future quantum communication networks, allowing an experimenter to determine which node of a quantum network receives information from a given source node.

**Optimal classical strategy**. Suppose that $A$, $B$, and $C$ are random variables with the same alphabet of size $d < \infty$. In this case, the fact that $X \in \{B, C\}$ is a causal intermediary for $A$ means that the map from $A$ to $X$ is a permutation. The first (second) causal hypothesis is that $B$ ($C$) is a permutation of $A$, while $C$ ($B$) is uniformly random. Other than this, no information about the functional relation between the variables is known to the experimenter. In particular, the experimenter does not know which permutation relates the variable $A$ to its causal intermediary $X$.

Let us determine how well one can distinguish between the two hypotheses with a finite number of experiments. In principle, we should examine all sequential strategies as in Fig. 1. However, in classical theory the problem can be greatly simplified: the optimal discrimination rate can be achieved by a parallel strategy, wherein the $N$ input variables are initially set to some prescribed set of values[31].

The possibility of an error arises is when the randomly fluctuating variable accidentally takes values that are compatible with a permutation, so that the outcome of the test gives no ground to discriminate between the two hypotheses. The probability of such inconclusive scenario is equal to $P(d, v)/d^N$, where $v$ is the number of distinct values of $A$ probed in the experiment and $P(d, v) = d!/(d - v)!$ is the number of injective functions from a $v$-element set to a $d$-element set. The probability of confusion is minimal for $v = 1$, leading to the overall error probability

$$p_{\mathrm{err}}^{\mathrm{C}} = \frac{1}{2d^{N-1}}. \qquad (1)$$

As a consequence, the rate at which the two causal hypotheses can be distinguished from each other is

$$R_{\mathrm{C}} = \log d. \qquad (2)$$

**A first quantum advantage**. Classical systems can be regarded as quantum systems that lost coherence across the states of a fixed basis, consisting of the classical states. But what if coherence is preserved? Could a coherent superposition of classical states be a better probe for the causal structure?

If the causal relations are restricted to reversible gates that permute the classical states, coherence offers an immediate advantage. The experimenter can prepare $N$ probes, each in the superposition $|e_0\rangle = \sum_{i=0}^{d-1} |i\rangle/\sqrt{d}$. Since the superposition is invariant under permutations, the unknown process will produce either $N$ copies of the state $|e_0\rangle\langle e_0|\otimes I/d$ or $N$ copies of the state $I/d\otimes|e_0\rangle\langle e_0|$, depending on which causal hypothesis holds. Using Helstrom's minimum error measurement[32], the error probability is reduced to

$$p_{\mathrm{err}}^{\mathrm{coh}} = \frac{1}{2d^N}. \qquad (3)$$

Compared with the classical error probability (1), the error probability of this simple quantum strategy is reduced by a factor $d$, which does not change the rate, but could be significant when the size of the alphabet is large.

Let us consider the full quantum version of the problem. Three quantum variables $A$, $B$, and $C$, corresponding to $d$-dimensional quantum systems, are promised to satisfy one of two causal hypotheses: either (i) the state of $B$ is obtained from the state of $A$ through an arbitrary unitary evolution and the state of $C$ is

maximally mixed, or (ii) the state of $C$ is obtained from the state of $A$ through an arbitrary unitary evolution and the state of $B$ is maximally mixed.

Despite the fact that now the cause–effect relation can be one of the infinitely many unitary gates, it turns out that the error probability (3) can still be attained. A universal quantum strategy, working for arbitrary unitary gates, is to prepare $d$ particles in the singlet state

$$|S_d\rangle = \frac{1}{\sqrt{d!}} \sum_{k_1,k_2,\cdots,k_d} \epsilon_{k_1 k_2 \ldots k_d} |k_1\rangle |k_2\rangle \cdots |k_d\rangle \quad (4)$$

where $\epsilon_{k_1 k_2 \ldots k_d}$ is the totally antisymmetric tensor and the sum ranges over all vectors in the computational basis. Then, each of the $d$ particles is used as an input to one use of the channel. Repeating the experiment for $t$ times, and performing Helstrom's minimum error measurement one can attain the error probability $p_{\mathrm{err}}^{\mathrm{coh}} = (2d^N)^{-1}$, with $N = td$, independently of the unitary gate representing the cause–effect relationship. In summary, the quantum error probability is at least $d$ times smaller than the best classical error probability, even if the cause–effect relationship is described by an arbitrary unitary gate.

**Optimality among simple parallel strategies.** We now show that the value (3) is optimal among all simple strategies where the unknown process is applied $N$ times in parallel on $N$ identical input systems, as in Fig. 3.

Optimality follows from a complementarity relation between the information about the causal structure and the information about the functional dependence between cause and effect. Suppose that the cause–effect dependence amounts to a unitary gate $U$ in some finite set U. The ability of a state $|\Psi\rangle$ to probe the cause–effect dependence can be quantified by the probability $p_{\mathrm{guess}}^{\mathrm{U}}$ of correctly guessing the unitary $U$ from the state $U^{\otimes N}|\Psi\rangle$. When the set of possibly unitaries has sufficient symmetry, we find that the probability of error in identifying the causal structure satisfies the lower bound

$$p_{\mathrm{err}} \geq \frac{1}{2d^N} \left\{ 1 + \frac{1}{2(d^N - 1)} \left( \frac{p_{\mathrm{guess}}^{\mathrm{U}} - \frac{1}{|\mathrm{U}|}}{\frac{1}{|\mathrm{U}|}} \right)^2 \right\} \quad (5)$$

(Supplementary Note 1). The higher the probability of success in guessing the cause–effect dependence, the higher the probability of error in identifying the causal structure. A consequence of the bound (5) is that the minimum error probability in identifying the causal intermediary is $(2d^N)^{-1}$, and is attained when the success probability $p_{\mathrm{guess}}^{\mathrm{U}}$ is equal to the random guess probability $1/|\mathrm{U}|$.

**Exponential reduction of the error probability.** The bound (5) shows that the discrimination rate of simple parallel strategies cannot exceed the classical discrimination rate $\log d$. We now show that that the rate can be doubled by entangling the $N$ probes with an additional reference system.

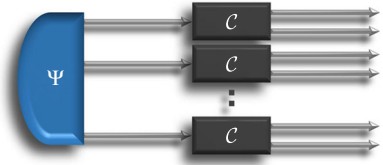

**Fig. 3** Simple parallel strategies. The unknown process $\mathcal{C}$ is probed for $N$ times, acting in parallel on $N$ identical systems, initially prepared in a correlated state $\Psi$

The working principle of our strategy is to build a quantum superposition of equivalent experimental setups. If no reference system is used, we know that the optimal strategy is to divide the $N$ probes into $N/d$ groups (assuming for simplicity that $N$ is a multiple of $d$), and to entangle the probes within each group. Clearly, different ways of dividing the $N$ inputs into groups of $d$ are equally optimal: it does not matter which particle is entangled with which, as long as all each particle is part of a singlet state. Still, we can imagine a machine that partitions the particles according to a certain configuration $i$ if a control system is in the state $|i\rangle$. When the control system is in a superposition, the machine will probe the unknown process in a superposition of configurations, as pictorially illustrated in Fig. 4. Explicitly, the optimal input state is

$$|\Psi\rangle = \frac{1}{\sqrt{G_{N,d}}} \sum_{i=1}^{G_{N,d}} \left( |S_d\rangle^{\otimes N/d} \right)_i \otimes |i\rangle, \quad (6)$$

where $i$ labels the different ways to partition $N$ identical objects into groups of $d$ elements, $G_{N,d}$ is the number of such ways, $\left( |S_d\rangle^{\otimes N/d} \right)_i$ is the product of $N/d$ singlet states arranged according to the $i$-th configuration, and $\{|i\rangle, i = 1, \ldots, G_{N,d}\}$ are orthogonal states of the reference system.

Classically, there would be no point in randomizing optimal configurations, because mixtures cannot reduce the error probability. But in the quantum case, the coherent superposition of equivalent configurations brings the error probability down to

$$p_{\mathrm{err}}^{\mathrm{Q}}(r) = \frac{r}{2d^N} \left( 1 - \sqrt{1 - r^{-2}} \right) \xrightarrow{r \gg 1} \frac{1}{4rd^N}, \quad (7)$$

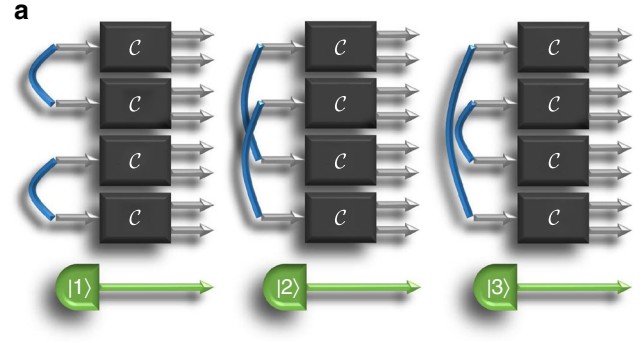

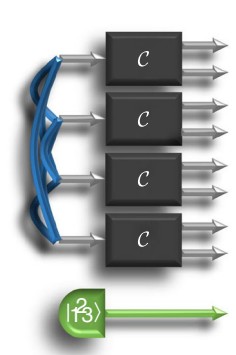

**Fig. 4** Coherent superposition of configurations. **a** shows the three different ways of dividing four quantum bits into groups of two. These three configurations are all equivalent for the identification of the causal intermediary. **b** pictorially illustrates a quantum superposition of configurations, with the choice of configuration correlated with the state of a control system

where $r$ is the number of linearly independent states of the form $\left(|S_d\rangle^{\otimes N/d}\right)_i$ (Supplementary Note 2).

To determine how much the error probability can be reduced, we only need to evaluate the number of linearly independent states. It turns out that this number grows as $d^N$, up to a polynomial factor (Supplementary Note 2 again). Taking the logarithm, we obtain the discrimination rate

$$R_Q = -\lim_{N\to\infty} \frac{\log p_{\text{err}}^Q}{N} = 2\log d, \qquad (8)$$

which is twice the classical discrimination rate (2). In fact, the asymptotic regime is already reached with a small number of interrogations, of the order of a few tens. For example, the causal relation between two quantum bits can be determined with an error probability smaller than $10^{-6}$ using with 12 interrogations, whereas 20 interrogations are necessary for classical binary variables.

The above strategy is universal, in that it applies to causal relationships described by arbitrary unitary gates. In particular, it applies to gates that permute the classical states. Hence, the ability to maintain coherence across the classical states and to generate entanglement with a reference system offers an exponential speedup with respect to the best classical strategy. In passing, we note that the universal quantum strategy is insensitive to the presence of perfectly correlated noise, such as the noise due to the lack of a reference frame[33], where each of the $N$ input variables is subjected to the same unknown unitary gate.

**The ultimate quantum limit**. So far, we examined strategies where the unknown process is applied in parallel to a large entangled state. Could a general sequence of interventions achieve an even better rate?

Finding the optimal sequential strategy is generally a hard problem. To address this problem, we introduce the fidelity divergence of two quantum channels $\mathcal{C}_1$ and $\mathcal{C}_2$, defined as

$$\partial F(\mathcal{C}_1, \mathcal{C}_2) = \inf_R \inf_{\rho_1, \rho_2} \frac{F[(\mathcal{C}_1 \otimes \mathcal{I}_R)(\rho_1), (\mathcal{C}_2 \otimes \mathcal{I}_R)(\rho_2)]}{F(\rho_1, \rho_2)}, \qquad (9)$$

where $\rho_1$ and $\rho_2$ are joint states of the channel's input and of the reference system $R$. It is understood that the infimum in the right-hand side is taken over pairs of states $(\rho_1, \rho_2)$ for which the fidelity $F(\rho_1, \rho_2)$ is non-zero, so that the expression on the right-hand side of Eq. (9) is well-defined.

The fidelity divergence quantifies the ability of channels $\mathcal{C}_1$ and $\mathcal{C}_2$ to move two states apart from each other. In the Methods section, we show that the error probability in distinguishing between $\mathcal{C}_1$ and $\mathcal{C}_2$ with $N$ queries is lower bounded as

$$p_{\text{err}}^{\text{seq}}(\mathcal{C}_1, \mathcal{C}_2; N) \geq \frac{\partial F(\mathcal{C}_1, \mathcal{C}_2)^N}{4}. \qquad (10)$$

In particular, suppose that the two channels $\mathcal{C}_1$ and $\mathcal{C}_2$ have the form $\mathcal{C}_1 = \mathcal{U} \otimes I/d$ and $\mathcal{C}_2 = I/d \otimes \mathcal{U}$, where $\mathcal{U}$ is a fixed unitary channel. In this case, we find that the fidelity divergence is $1/d^2$. Hence, the error probability satisfies the bound

$$p_{\text{err}}^{\text{seq}}(\mathcal{C}_1, \mathcal{C}_2; N) \geq \frac{1}{4d^{2N}}. \qquad (11)$$

In the causal intermediary problem, the unitary gate $\mathcal{U}$ is unknown, and therefore the error probability can only be larger than $p_{\text{err}}^{\text{seq}}(\mathcal{C}_1, \mathcal{C}_2; N)$. Hence, the identification of the causal intermediary cannot occur at a rate faster than $2\log d$.

Equation (11) limits all sequential quantum strategies. But in fact quantum theory is also compatible with scenarios where physical processes take place in an indefinite order[23,24]. Could the

rate be increased if the experimenter had access to exotic phenomena involving indefinite order?

The answer is negative. In the Methods section, we develop the concepts and methods needed to answer this question, and we show that the minimum error probability in distinguishing between the two channels $\mathcal{C}_1 = \mathcal{I} \otimes I/d$ and $\mathcal{C}_2 = I/d \otimes \mathcal{I}$ using arbitrary setups with indefinite order satisfies the bound

$$p_{\text{err}}^{\text{ind}}(\mathcal{C}_1, \mathcal{C}_2; N) \geq \frac{1 - \sqrt{1 - \frac{1}{d^{2N}}}}{2}. \qquad (12)$$

Clearly, this bound applies to the causal intermediary problem, which is harder than the discrimination of the two specific channels $\mathcal{C}_1 = \mathcal{I} \otimes I/d$ and $\mathcal{C}_2 = I/d \otimes \mathcal{I}$. Hence, the rate $R_Q = 2\log d$ represents the ultimate quantum limit to the identification of a causal intermediary.

**Extension to arbitrary numbers of hypotheses**. The quantum advantage demonstrated in the previous sections can be extended to the identification of the causal intermediary among an arbitrary number $k$ of candidate variables. The best classical strategy still consists of initializing all variables to the same value. Errors arise when the values of two or more output variables are compatible with an invertible function. In the limit of many repetitions, the minimum error probability is $p_{\text{err,k}}^C = (k-1)/(2d^{N-1}) + O(d^{-2N})$. (Supplementary Note 3). For quantum strategies, the best option among simple parallel strategies is still to divide the input particles into $N/d$ groups of $d$ particles and to initialize each group in the singlet state. In Supplementary Note 4, we show that this strategy reduces the error probability to $p_{\text{err,k}}^{\text{coh}} = (k-1)/(2d^N) + O(d^{-2N})$, for causal relations represented by arbitrary unitary gates.

An exponentially smaller error probability can be achieved using the input state (6). The evaluation of the error probability is more complex than in the two-hypothesis case, but the end result is the same: when the causal dependency is probed $N$ times, the quantum error probability decays at the exponential rate $R_Q = 2\log d$, twice the rate of the best classical strategy (see Supplementary Note 5 for the technical details).

**Applications to other tests of causal hypotheses**. The strategies developed in the previous sections can be applied to the identification of causal relations in a variety of scenarios. For example, they can be used to decide whether there is a causal link between two variables $A$ and $B$. More specifically, they can be used to determine whether variable $B$ is a causal intermediary for variable $A$ or whether $B$ fluctuates at random independently of $A$. Also in this case, the error probability of the best classical strategy is $1/(2d^{N-1})$, whereas preparing $N/d$ copies of the singlet yields error probability $1/(2d^N)$.

By superposing all possible partitions of the $N$ inputs into groups of $d$, one can boost the discrimination rate from $\log d$ to $2\log d$. One could speculate that, in the future, such a fast identification could be useful as a quantum version of the ping protocol, capable of establishing whether there exists a quantum communication link between two nodes of a quantum internet[34].

Another application of our techniques is in the problem of identifying the cause of a given variable. Suppose that one of $k$ variables $A_1, A_2, \ldots, A_k$ is the cause for a given variable $B$. An example of this situation arises in genetics, when trying to identify the gene responsible for a certain characteristic. Here, the interesting scenario is when the number of candidate causes is large.

Classically, the problem is to find the variable $A_x$ such that $B$ is a function of $A_x$. For simplicity, we first assume that all variables

have the same $d$-dimensional alphabet, and that the function from $A_x$ to $B$ is the identity, namely $b = a_x$. In this case, the cause can be identified without any error by probing the unknown process for $\lceil \log_d k \rceil$ times. The identification is done by a simple search algorithm, where one divides the candidate variables in $d$ groups and initializes the input variables in the $i$-th group to the value $i$. In this way, $d - 1$ groups can be ruled out, and one can iterate the search in the remaining group. Using a decision tree argument[35], it is not hard to see that $\lceil \log_d k \rceil$ is the minimum number of queries needed to identify the unknown process in the worst case scenario.

In the quantum version of the problem, we find that the number of queries can be cut down by approximately a half when the number of hypotheses is large. The trick is to prepare $k$ maximally entangled states, and to apply the unknown process to the first system of each pair. Repeating this procedure for $N$ times and using results on port-based teleportation[36] we find that the error probability is $p_{err} = (k - 1)/(d^{2N} + k - 1)$. Hence, $N = \lceil (1 + \epsilon)(\log_d k)/2 \rceil$ queries are sufficient to identify the cause with vanishing error probability in the large $k$ limit.

In Supplementary Note 6, we consider the more complex scenario where the functional dependence between the cause and effect is unknown, and the only assumption is that the effect is a causal intermediary of the cause. Despite the lack of information about the functional dependence, we show that the correct cause can be still identified with high probability using $N = \lceil (1 + \epsilon)(\log_d k)/2 \rceil$ calls to the unknown process. The fast identification of the cause is achieved by dividing the $N$ copies of each input variable $A_i$ into groups of $d$ copies, preparing each group in the singlet state, and entangling the configuration of the groupings with an external reference system. Once again, the superposition of multiple equivalent setups leads to a quantum speedup over the best classical strategy.

## Discussion

We showed that quantum mechanics enhances our ability to detect direct cause–effect links. This finding motivates the exploration of more complex networks of causal relations, including intermediate nodes and global causal dependences between groups of variables[1–3]. The development of new techniques for testing causal relations could find applications to future quantum communication networks, providing a fast way to test the presence of communication links. It could also assist the design of intelligent quantum machines, in a similar way as classical causal discovery algorithms have been useful in classical artificial intelligence. In view of such applications, it is important to go beyond the noiseless scenario considered in this paper, and to address scenarios where the cause–effect relationships are obfuscated by noise. The techniques developed in our work already provide some insights in this direction. Quite interestingly, one can show that the quantum advantage persists in the presence of depolarizing noise, provided that the noise level is not too high (see Supplementary Note 7). A complete study of the noisy scenario, however, remains an open direction of future research.

Another direction of future investigation is foundational. Given the advantage of quantum theory over classical theory, it is tempting to ask whether alternative physical theories could offer even larger advantages. Interesting candidates are theories that admit more powerful dense coding protocols than quantum theory[37], as one might expect super-quantum advantages to arise from the presence of stronger correlations with the reference system. In a similar vein, one could explore physical theories with higher dimensional state spaces, such as Zyczkowski's quartic theory[38], or quantum theory on quaternionic Hilbert spaces[39]. Indeed, it is intriguing to observe that the classical rate $R^C = \log d$ and the quantum rate $R^Q = 2 \log d$ are equal to the logarithms of the dimensions of the classical and quantum state spaces, respectively. In general, one may expect a relationship between the dimension of the state space and the rate. Should super-quantum advantages emerge, it would be natural to ask which physical principle determines the causal identification power of quantum mechanics. An intriguing possibility is that one of the hidden physical principles of quantum theory could be a principle on the ability to distinguish alternative causal hypotheses.

## Methods

**Properties of the fidelity divergence.** Here, we derive two properties of the fidelity divergence defined in Eq. (9). First, the fidelity divergence provides a lower bound on the probability of misidentifying a channel with another:

*Proposition 1* The probability of error in distinguishing between two quantum channels $\mathcal{C}_1$ and $\mathcal{C}_2$ with $N$ queries is lower bounded as $p_{err}^{seq}(\mathcal{C}_1, \mathcal{C}_2; N) \geq \partial F(\mathcal{C}_1, \mathcal{C}_2)^N/4$.

The bound can be obtained in the following way. Let $\rho_x^{(N)}$ be the output state of a circuit as in Fig. 1. Then, we have the bound

$$\begin{aligned} p_{err}^{seq}(\mathcal{C}_1, \mathcal{C}_2; N) &= \tfrac{1}{2}\left(1 - \tfrac{1}{2}\left\|\rho_1^{(N)} - \rho_2^{(N)}\right\|_1\right) \\ &\geq \tfrac{1}{2}\left(1 - \sqrt{1 - F(\rho_1^{(N)}, \rho_2^{(N)})}\right) \\ &\geq \tfrac{1}{2}\left[1 - \sqrt{1 - \partial F^N(\mathcal{C}_1, \mathcal{C}_2)}\right] \\ &\geq \tfrac{1}{2}\left[1 - \left(1 - \tfrac{\partial F^N(\mathcal{C}_1, \mathcal{C}_2)}{2}\right)\right] \\ &= \tfrac{\partial F(\mathcal{C}_1, \mathcal{C}_2)^N}{4}. \end{aligned} \qquad (13)$$

The first line follows from Helstrom's theorem[32], and the second line follows from the Fuchs–Van De Graaf Inequality[40]. The third line follows from the definition of the fidelity divergence (9), which implies that the fidelity between the states right after the $(t + 1)$-th use of the unknown channel $\mathcal{C}_x$, denoted by $\rho_{x,t+1}$, satisfies the bound

$$\begin{aligned} F(\rho_{1,t+1}, \rho_{2,t+1}) &\geq \partial F(\mathcal{C}_1, \mathcal{C}_2) F(\mathcal{U}_{t+1}\rho_{1,t}, \mathcal{U}_{t+1}\rho_{2,t}) \\ &\geq \partial F(\mathcal{C}_1, \mathcal{C}_2) F(\rho_{1,t}, \rho_{2,t}), \end{aligned} \qquad (14)$$

where $\mathcal{U}_{t+1}$ is the $(t + 1)$-th operation in Fig. 1. The fourth line follows from the elementary inequality $\sqrt{1 - t} \leq 1 - t/2$.

Another important property is that the fidelity divergence can be evaluated on pure states. The proof is simple: let $\rho_1$ and $\rho_2$ be two arbitrary states of the composite system $AR$, where $R$ is an arbitrary reference system. By Uhlmann's theorem[41], there exists a third system $E$ and two purifications $|\Psi_1\rangle, |\Psi_2\rangle \in \mathcal{H}_A \otimes \mathcal{H}_R \otimes \mathcal{H}_E$, such that $F(\Psi_1, \Psi_2) = F(\rho_1, \rho_2)$. On the other hand, the monotonicity of the fidelity under partial trace[42], ensures that the fidelity between the output states $(\mathcal{C}_1 \otimes \mathcal{I}_{RE})(\Psi_1)$ and $(\mathcal{C}_2 \otimes \mathcal{I}_{RE})(\Psi_2)$ cannot be larger than the fidelity between the states $(\mathcal{C}_1 \otimes \mathcal{I}_R)(\rho_1)$ and $(\mathcal{C}_2 \otimes \mathcal{I}_R)(\rho_2)$. Hence, the minimization on the right-hand side of Eq. (9) can be restricted without loss of generality to pure states.

**Fidelity divergence for the identification of the causal intermediary.** Let us see how the fidelity divergence can be applied to our causal identification problem. The two channels are of the form $\mathcal{C}_{1,U}(\rho) = U\rho U^\dagger \otimes I/d$ and $\mathcal{C}_{2,V} = I/d \otimes V\rho V^\dagger$, where $U$ and $V$ are two unknown unitary gates. Since we are interested in the worst case scenario, every choice of $U$ and $V$ will give an upper bound to the discrimination rate. In particular, we pick $U = V$.

*Proposition 2* The fidelity divergence for the two channels $\mathcal{C}_{1,U}$ and $\mathcal{C}_{2,U}$ is $\partial F(\mathcal{C}_{1,U}, \mathcal{C}_{2,U}) = 1/d^2$.

By the unitary invariance of the fidelity, $\partial F(\mathcal{C}_{1,U}, \mathcal{C}_{2,U})$ is independent of $U$. Without loss of generality, let us pick $U = I$. For a generic reference system $R$ and two generic pure states $|\Psi_1\rangle, |\Psi_2\rangle \in \mathcal{H}_A \otimes \mathcal{H}_R$, the two output states are

$$\begin{aligned} \rho_1' &= (\mathcal{C}_{1,I} \otimes \mathcal{I}_R)(\Psi_1) = (\Psi_1)_{BR} \otimes \tfrac{I_C}{d} \\ \rho_2' &= (\mathcal{C}_{2,I} \otimes \mathcal{I}_R)(\Psi_2) = \tfrac{I_B}{d} \otimes (\Psi_1)_{CR}, \end{aligned} \qquad (15)$$

up to reordering of the Hilbert spaces. The fidelity can be computed with the relation

$$F(\rho_1', \rho_2') = \frac{\left|\mathrm{Tr}\left[\sqrt{(\Psi_1)_{BR}(\Psi_2)_{CR}(\Psi_1)_{BR}}\right]\right|^2}{d^2}, \qquad (16)$$

where we omitted the identity operators for the sake of brevity. Let us expand the

input states as

$$|\Psi_x\rangle = \sum_n |\phi_{xn}\rangle \otimes |n\rangle, \qquad x \in \{0, 1\} \tag{17}$$

where $\{|n\rangle\}$ is an orthonormal basis for the reference system, and $\{|\phi_{xn}\rangle\}$ is a set of unnormalized vectors. Inserting Eq. (17) into Eq. (16), we obtain the expression

$$F(\rho_1', \rho_2') = \frac{\left| \mathrm{Tr}\left[ \sqrt{C^\dagger C} \right] \right|^2}{d^2} = \frac{|Tr|C||^2}{d^2}, \tag{18}$$

with $C = \sum_n |\phi_{1n}\rangle\langle\phi_{2n}|$. On the other hand, the fidelity between the input states is

$$F(\rho_1, \rho_2) = |\langle\Psi_1|\Psi_2\rangle|^2 = |\mathrm{Tr}[C]|^2. \tag{19}$$

Hence, the fidelity divergence satisfies the bound

$$\begin{aligned}
\partial F(\mathcal{C}_1, \mathcal{C}_2) &= \inf_R \inf_{\rho_1, \rho_2} \frac{F(\rho_1', \rho_2')}{F(\rho_1, \rho_2)} \\
&= \frac{1}{d^2} \inf_C \left| \frac{\mathrm{Tr}|C|}{\mathrm{Tr}[C]} \right|^2 \\
&\geq \frac{1}{d^2},
\end{aligned} \tag{20}$$

having used the inequality $|\mathrm{Tr}[C]| \leq \mathrm{Tr}|C|$, valid for every operator $C$. The inequality holds with the equality sign whenever $C$ is positive. This condition is satisfied, e.g., when the input states $|\Psi_1\rangle$ and $|\Psi_2\rangle$ are identical.

**Quantum strategies with indefinite causal order.** In principle, quantum mechanics is compatible with situations where multiple processes are combined in indefinite order[23,24]. This suggests that an experimenter could devise new ways to probe quantum channels, allowing the relative order among different uses of the same channel to be indefinite. We call such strategies indefinite testers.

Consider the problem of identifying a channel $\mathcal{C}_x$ from $N$ uses. The input resource is the channel $\mathcal{C}_x^{\otimes N}$, representing $N$ identical black boxes that can be arranged in any desired order. Besides the product of $N$ independent channels, the most general class of channels with this property is the class of no-signaling channels with $N$ pairs of input/output systems.

Mathematically, an indefinite tester is a linear map from the set of no-signaling channels to the set of probability distributions over a given set of outcomes. Equivalently, the tester can be described by a set of operators $\{T_x\}$, where each operator $T_x$ acts on the Hilbert space $\otimes_i (\mathcal{H}_i^{\mathrm{in}} \otimes \mathcal{H}_i^{\mathrm{out}})$, where $\mathcal{H}_i^{\mathrm{in}}$ and $\mathcal{H}_i^{\mathrm{out}}$ are the Hilbert spaces of the input and output system in the $i$-th pair, respectively. When the test is performed on a no-signaling channel $\mathcal{C}$, the probability of the outcome $x$ is given by the generalized Born rule $p_x = \mathrm{Tr}[T_x C]$, where $C$ is the Choi operator of the channel $\mathcal{C}$[43]. The normalization of the probabilities

$$\sum_x \mathrm{Tr}[T_x C] = 1 \tag{21}$$

is required to hold for every no-signaling channel $\mathcal{C}$.

Consider the problem of distinguishing between a set of no-signaling channels $\{\mathcal{C}_x\}$ using an indefinite tester. For every probability distribution $\{\pi_x\}$, the worst-case probability of error satisfies the bound

$$p_{\mathrm{err}}^{\mathrm{ind}} \geq 1 - \sum_x \pi_x \mathrm{Tr}[T_x C_x]. \tag{22}$$

Now, suppose that there exists a constant $\lambda$ and a no-signaling channel $\mathcal{C}$ such that

$$\lambda C \geq \pi_x C_x \tag{23}$$

for every $x$. Substituting Eq. (23) into Eq. (22) one obtains the bound

$$p_{\mathrm{err}}^{\mathrm{ind}} \geq 1 - \lambda \sum_x \mathrm{Tr}[T_x C] = 1 - \lambda, \tag{24}$$

having used the normalization condition (21). The bound (24) can be seen as a generalization of the classical Yuen–Kennedy–Lax bound for quantum state discrimination[44].

We now apply the bound (24) to the task of distinguishing between the two channels $\mathcal{C}_{1,I} = (\mathcal{U} \otimes I/d)^{\otimes N}$ and $\mathcal{C}_{2,I} = (I/d \otimes \mathcal{U})^{\otimes N}$. To this purpose, we consider the universal cloning channel[45]

$$\mathcal{C}_\pm := \frac{2}{d^N + 1} P_+ (\rho \otimes I^{\otimes N}) P_+, \tag{25}$$

and the universal NOT channel[46]

$$\mathcal{C}_\pm := \frac{2}{d^N - 1} P_- (\rho \otimes I^{\otimes N}) P_-, \tag{26}$$

with $P_\pm = (I \pm \mathrm{SWAP})/2$, and SWAP being the unitary operator that swaps between the even and odd output spaces. It is easy to verify that both channels are no-signaling. Moreover, we find that the convex combination $\mathcal{C} = p_+ \mathcal{C}_+ + p_- \mathcal{C}_-$ with $p_\pm = \sqrt{\frac{d^N \pm 1}{2d^N}} / \left( \sqrt{\frac{d^N+1}{2d^N}} + \sqrt{\frac{d^N-1}{2d^N}} \right)$ satisfies the condition (23) with $\lambda = \frac{1}{2} \left( \sqrt{\frac{d^N+1}{2d^N}} + \sqrt{\frac{d^N-1}{2d^N}} \right)^2$ (see Supplementary Note 8 for technical details). Hence, the

bound (24) becomes

$$p_{\mathrm{err}}^{\mathrm{ind}} \geq 1 - \lambda = \frac{1 - \sqrt{1 - \frac{1}{d^{2N}}}}{2} \geq \frac{1}{4d^{2N}}. \tag{27}$$

The above bound implies that the discrimination rate of quantum strategies with indefinite order cannot exceed $2 \log d$.

## Data availability
The authors declare that the data supporting the findings of this study are available within the paper and in the Supplementary Information files.

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

## Acknowledgements

The authors acknowledge Robert Spekkens, David Schmidt, Lucien Hardy, Sergii Strelchuk, Akihito Soeda, and Thomas Gonda for stimulating discussions. This work is supported by the National Natural Science Foundation of China through Grant 11675136, the Croucher Foundation, John Templeton Foundation, Project 60609, Quantum Causal Structures, the Canadian Institute for Advanced Research (CIFAR), the Hong Research Grant Council through Grants 17300317 and 17300918, and the Foundational Questions Institute through Grant FQXi-RFP3-1325. This publication was made possible through the support of a grant from the John Templeton Foundation. The opinions expressed in this publication are those of the authors and do not necessarily reflect the views of the John Templeton Foundation. This research was supported in part by Perimeter Institute for Theoretical Physics. Research at Perimeter Institute is supported by the Government of Canada through the Department of Innovation, Science and Economic Development Canada and by the Province of Ontario through the Ministry of Research, Innovation and Science.

## Author contributions

Both the authors contributed substantially to the research presented in this paper and to the preparation of the manuscript.

## Additional information

**Competing interests:** The authors declare no competing interests.

