## [Peer Review File · Nature Communications]

Reviewers' comments:

Reviewer #1 (Remarks to the Author):

The paper considers the problem of distinguishing between two "causal hypotheses": namely, given 3 variables A, B and C where A is either the cause of B or of C, one has to decide which of the two cases (A causes B or A causes C) is the correct one.

The authors consider different scenarios to solve the problem, either classically or quantumly, with or without a reference system, by probing the variables in parallel or in series. They show that quantum mechanics can provide an exponential decrease for the probability of error, compared to the classical case.

The research is quite original. The study of causal structures in quantum mechanics has attracted a lot of interest recently, and the authors exhibit here a convincing example where quantum mechanics provides an exponential advantage of a new kind over classical physics. The paper is thus quite timely. It is clearly written, and should be accessible and of interest for a wide audience.

Before I can recommend the paper for publication, however, I have a number of remarks that I would encourage the authors to consider.

*** In the main text:

- top left of p.2, in "...where variable A may cause either variable B or variable C".

At this point it is not clear what "a variable causes another" means. Later in the paper the problem is formalised in terms of identifying "causal intermediaries". Is it precisely what is meant on the top left of p.2? This may be clarified (or at least it should be said there that the problem will be rigorously formalised later.).

- bottom left of p.2: why consider the worst case error probability over all processes, rather than the average?

It is claimed in the Suppl. Mat. that considering the average would actually give the same error probabilities; this could be mentioned in the main text.

- Eq. (1): why is the rate defined like this (and why is it called a "rate")? Is that a standard definition (in which case some reference may be relevant)? How is the operational meaning justified?

- Before Eq. (2), the process $\{\text{cal } C\}$ should be defined. (Note that $\{\text{cal } C\}$ was denoting before a channel from A to B and C; here it is a channel from A to B only.)

- p.3, beginning of the "Optimal classical strategy" section: what exactly is meant by "identical random variables"? That they have the same alphabet? Or the same probability distribution? Or that they are perfectly correlated? This should be clarified.

- bottom left of p.3, the notation N_0 , N_1 , etc is introduced, but does not seem to be used anywhere?

- In the comparison between classical and quantum strategies, the authors compare classical variables with an alphabet of size d to quantum systems of Hilbert space dimension d .

Why is this comparison appropriate? This should certainly be justified somewhere!

Note that if one one comparing the classical alphabet size to the dimension of the quantum state space (d^2) instead of the Hilbert space dimension (d), the quantum advantage would disappear.

Why would this not be an appropriate comparison?

- top left of p.4, when referring to Figure 3: Fig. 3 does in fact not illustrate the 3 types of strategies, but only parallel ones.

- After Eq. (7), some intuitive explanation is given for the origin of the quantum advantage in terms of the "complementarity between the information about the causal structure and the information about the functional dependence between cause and effect".

I was wondering if this was just an intuition, or can it be put of firm grounds, e.g. by properly quantifying this complementarity?

- In Eq. (9): does the notation 'r' refer to the same quantity as the notation 'm_lambda' in the Suppl. Mat.? For clarity one could maybe use only one notation?

- Right after Eq. (9): shouldn't one refer to Suppl. Note 4, rather than 3?

- When referring to the sequential scheme, the authors talk about "the ultimate limit" set by quantum mechanics. Nevertheless, they mention in their conclusion the intriguing question, whether one could further decrease the error probability by using processes with indefinite causal order. If that was the case (with an implementable process, like the "quantum switch"), couldn't be the "ultimate limit" beaten by quantum mechanics? I would then suggest to avoid claiming that they already obtain "the ultimate limit" (or to talk about "the most general test" in the SM).

- In the second part of the Methods, the authors set $U = V = I$, as this is enough to obtain an upper bound to the discrimination rate in the worst case scenario. However, their goal there is to calculate the fidelity divergence for 2 channels (Proposition 2). Is it then also justified for this calculation to set $U = V = I$? Why?

*** In the Suppl. Mat.:

- Eq. (4), shouldn't it be 'max'?

- 1st line of p.2, the number of the section the authors refer to is missing.

- 2nd line of p.2: should 'Hence' be understood as 'It turns out that...'?

- No proof is given for Lemma 2; is that a similar simple adaptation of Holevo's argument?

- 3rd line of Suppl. Note 2: 'n' should be a capital 'N'.

- In Lemmas 3 and 4, it may be useful to recall that one assumes N is a multiple of d .

- Eq. (12): does one really need to consider $T_{\text{out}}^{(N)}$ in the argument?

To reach the conclusion of Eq. (14), doesn't one also need some convexity argument?

- In Eq. (46) (and then in Lemma 8) I found $c(N) = \prod_{i=1}^d (d-i)!$, i.e., a fixed value independent of N , that does not tend to 1 as N grows to infinity (but remains a constant).

This should be checked again.

If I am wrong and c indeed depends on N , one shouldn't talk about "a constant tending to 1".

- In Lemma 6, by "Let A be the cause of exactly one variable", do the authors mean it has only one causal intermediary? This may be clarified.

- Beginning of the proof of Lemma 6: why is there no loss of generality in considering strategies

with all input variables initialised to the same value? This should be justified.

- Before Eqs. (61) and (63), I guess it should be " $\lambda_{j/l} \neq \lambda_0$ ", rather than " $\neq 0$ "?

Reviewer #2 (Remarks to the Author):

In this paper, the authors consider the task of distinguishing different causal relationships, given finite data. They demonstrate that quantum physics allows an exponential reduction of the probability of error, relative to classical physics, for a given number of trials. Specifically, they consider two possible causal relationships among three variables A, B, and C: Either A is a cause of B but not of C, or A is a cause of C but not of B. The variable for which A is not a cause is assumed to be uniformly random.

While the specific problem considered might seem slightly artificial (and the authors do not provide any example of an application - is there one?), the novel demonstration of an exponential quantum advantage in causal inference is very interesting and certainly deserves high visibility and broad dissemination. Causal inference is essential in many branches of science, as pointed out by the authors, as well as a field of research in itself. And the study of causality in the quantum regime is currently highly active. This work is therefore likely to have significant and broad impact, and to spark further work on quantum causal inference.

The manuscript is well written and the analysis is thorough, although with most of the mathematical details relegated to the supplementary material. The derivations are sound, as far as I can tell, although I have not gone through all of them in detail (in particular, I am not familiar with the representations in terms of Young diagrams employed in the supplementary note 3 and onwards), and the techniques quite novel in the context of causal inference, I believe.

I recommend publication in Nature Communications.

I do have a couple of questions, which I would ask the authors to clarify, and a few minor comments:

1)

In the problem considered by the authors, it is assumed that the causal intermediary (B or C) of A is related to A by an invertible process - i.e. B or C retains complete information about A. However, it is easy to imagine that this might not be the case in a realistic situation. Can the authors say anything about what happens when the causal intermediary contains only partial information about A? Clearly, in the limit where no information is retained, quantum and classical strategies must perform equally bad. Is there still a quantum advantage in the regime in between?

2)

Below Eq. (9), the authors explain that the number of linearly independent states entering in the sum in Eq. (8) grows d^N . I think this claim is based on the argument around Eqs. (45)-(47) in the supplementary material. However, it is not obvious to me why the statement about $m_{\{\lambda_0\}}$ proven there implies the claim in the main text. Could the authors explain this?

3)

In the review of previous work in the introduction and the discussion of interventions, it would appear relevant to reference work showing that the quantum behaviour under interventions can be very different than the classical one, for fixed scenarios. E.g.

- Chaves, Carvacho, Agresti, Di Giulio, Aolita, Giacomini, Sciarrino, Nature Physics 14, 291 (2018).
- Van Himbeeck, Brask, Pironio, Ramanathan, Sainz, Wolfe, arXiv:1804.04119v1 [quant-ph] (2018).

4)

On p. 4 the authors write that "The three type of strategies corresponding to the above three features are illustrated in Figure 3". However, the third strategy (coherence in time) doesn't seem to be illustrated in Fig. 3?

5)

Below Eq. (9), I think N in the exponent should be N/d ?

6)

The formulation in the outlook that quantum physics may be "at least, never worse" than classical physics is perhaps a bit confusing, as, for many purposes, classical physics is contained in quantum physics as a special case. In that case, obviously quantum can be no worse than classical. I guess the authors maybe refer to the challenges when genuinely quantum causal structures need to be identified. A more precise formulation might be clearer here.

7)

In Lemma 1 on p.2 of the supplementary material, I find the notation a bit strange. The authors refer to the worst-case error probability for distinguishing channels $C_{\{1,V\}}$ and $C_{\{2,U\}}$. But the worst-case error probability refers to the minimisation over U and V (as in Eq. (4)), and so does not depend on U or V . Neither does the average error probability. So perhaps in the lemma it would be better to leave U and V out of the subscripts.

8)

Below Eq. (38) in the supplementary material, perhaps it would be clearer to say that $m_{\{\lambda_0\}}$ has to be exponentially large rather than "asymptotically" large.

9)

In the Methods section, the authors explain that the third line in Eq. (16) follows from the definition of the fidelity divergence. However, this was not obvious to me. Could the authors elaborate?

Reviewer #3 (Remarks to the Author):

Dear Editors,
Dear Authors,

First of all I would like to write that I have read submitted manuscript 'Quantum speedup in testing casual hypotheses' by Giulio Chiribella and Daniel Ebler with the great interest.

Authors in their manuscript prove that quantum physics offers an exponential advantage over classical one in the task of testing casual hypotheses. Namely, the main result states that quantum error probability of guessing incorrect hypothesis is d times smaller than in the classical counterpart. In all proofs Authors extensively use tools coming from representation theory of $SU(d)$ and $S(n)$, in particular famous Schur-Weyl duality. Symmetries occurring in the problem allowed to find analytical expressions for optimal states and error probabilities in four different scenarios. I have to emphasize here that whenever we ask about optimal quantities in some protocols, obtaining answer in closed form, in other words being able to rid of optimisation over all

parameters etc. is

a very hard task. In my opinion all proofs in appendices are correct. They are written clearly and explicitly, allowing potential Reader to follow all key steps in the argumentation.

Additionally, I have a few minor remarks (according <https://arxiv.org/abs/1806.06459>):

1) In my opinion there is no deep motivation for considering such kind of problem(s) through whole introductory section. Of course, I am aware of the main goal is better understanding of surrounding World and di

ferences between quantum and classical regimes. But for such high-impact journal we would expect much more deeper motivation and implications, rather than some general remarks and conjectures. What exactly factor δ offers to us? It is well known that in many situations quantumness offers more than classicality, and I would like to know why, in particular this result deserves for publication in Nature Communication.

2) Full stop is missing in the first paragraph on the page 4. Just before the sentence 'The three type of strategies...'

3) In the fourth line of equation (16) it should be ∂ instead of δ .

4) It should be $\phi_{\{x_n\}}$ instead of $\psi_{\{x_n\}}$ in the text, between equations (19) and (20).

5) In the sentence below equation (57) the reference is needed.

6) There is missing comma just after expression in equation (59).

7) Full stop just after expression in equation (113) is unnecessary.

8) For me considerations in Appendix G look similar to probabilistic port-based teleportation (in the context of signals discrimination). But somehow probability of success behaves differently.

Could Authors elaborate on this?

Results presented in the manuscript are interesting and correct, whole paper is a nice and very solid piece of work, but in this form probably it will not be interesting for broader audience, even in the field of quantum information theory. I strongly recommend to include more detailed discussion on implications of obtained results or find other, specialised journal for the publication. Having additional results I would be happy to reconsider manuscript for the publication in Nature Communication.

List of changes

The revised manuscript contains improvements both in the presentation and in the content, which feature new results stimulated by the Referees' comments.

The main changes are as follows:

(1) To facilitate the comparison between classical and quantum strategies, we included a discussion of the quantum case where the cause-effect relationship is induced by a permutation of basis states. This discussion can be found around Equation (5) of the main text.

(2) A complementarity relation between tests of the causal structure and tests of the functional dependence between cause and effect has been included. The complementarity relation provides a new proof of optimality for our quantum strategy without reference system. The new proof is provided in Supplementary Note 1.

(3) We proved that the rate of our quantum strategy is optimal even among strategies using indefinite causal order. The discussion is provided on page 5. The proof is provided in the end of the supplemental material, together with a brief outline of the framework of "indefinite testers" used to probe quantum processes in an indefinite order, and with the derivation of a Yuen-Kennedy-Lax bound for discrimination of processes in an indefinite order.

(4) We extended our analysis of the quantum advantage to two new tasks, including: (a) the detection of a causal link between two variables, and (b) the identification of the cause of a given effect. The corresponding discussion can be found in the left column of page 6.

(5) Abstract, introduction, and conclusions have been modified to take into account the new results.

(6) The presentation has been improved and the supplemental material streamlined.

(7) New references have been added to reflect additions in the main text and supplemental material.

Response to Reviewer #1

Referee: The paper considers the problem of distinguishing between two “causal hypotheses”: namely, given 3 variables A, B and C where A is either the cause of B or of C, one has to decide which of the two cases (A causes B or A causes C) is the correct one.

The authors consider different scenarios to solve the problem, either classically or quantumly, with or without a reference system, by probing the variables in parallel or in series. They show that quantum mechanics can provide an exponential decrease for the probability of error, compared to the classical case.

The research is quite original. The study of causal structures in quantum mechanics has attracted a lot of interest recently, and the authors exhibit here a convincing example where quantum mechanics provides an exponential advantage of a new kind over classical physics. The paper is thus quite timely. It is clearly written, and should be accessible and of interest for a wide audience.

Authors: Thank you for the positive assessment of the broad interest, soundness, and timeliness of our work.

Referee: Before I can recommend the paper for publication, however, I have a number of remarks that I would encourage the authors to consider.

- top left of p.2, in “...where variable A may cause either variable B or variable C”. At this point it is not clear what “a variable causes another” means. Later in the paper the problem is formalised in terms of identifying “causal intermediaries”. Is it precisely what is meant on the top left of p.2? This may be clarified (or at least it should be said there that the problem will be rigorously formalised later.).

Authors: Thanks for suggesting this clarification. Actually, we did not refer to causal intermediaries at that point. Instead, we simply meant that the map from A to B is non-constant. In the revised version we clarified this point.

Referee:- bottom left of p.2: why consider the worst case error probability over all processes, rather than the average?

Authors: The worst-case error probability is a stronger performance guarantee. For example, there could be situations where the average error is small, but for some specific processes the error is large, meaning that for some processes the estimating strategy is unreliable. Another reason for preferring the worst-case approach is that it bypasses the problem of choosing a prior for the unknown processes.

Referee: It is claimed in the Suppl. Mat. that considering the average would actually give the same error probabilities; this could be mentioned in the main

text.

Authors: It is important to stress that the equality between worst-case and average does not hold in general, but only when the set of unitaries form a group. Following up on your comment, in the main text we emphasized that the error probabilities of our strategies are independent of the unitary gates. This implies in particular that the average error probability is equal to the worst-case error probability.

Referee: - Eq. (1): why is the rate defined like this (and why is it called a "rate")? Is that a standard definition (in which case some reference may be relevant)? How is the operational meaning justified?

Authors: The definition of rate is indeed standard, see e.g. the works on Stein's lemma or on the Chernoff bound. The idea is that the probability of error typically decays exponentially with the number of experiments, and the constant in the exponential determines the decay rate. Given the exponential scaling $p_{\text{err}} \approx 2^{-RN}$ and given an error threshold ϵ , the number of samples of the unknown process grows as $N \approx \log(1/\epsilon)/R$. In the revised version, the above comments have been incorporated in the main text.

Referee: - Before Eq. (2), the process $\{\mathcal{C}\}$ should be defined. (Note that $\{\mathcal{C}\}$ was denoting before a channel from A to B and C; here it is a channel from A to B only.)

Authors: Agree; clarification added.

Referee: - p.3, beginning of the "Optimal classical strategy" section: what exactly is meant by "identical random variables"? That they have the same alphabet? Or the same probability distribution? Or that they are perfectly correlated? This should be clarified.

Authors: Same alphabet; clarification added.

Referee: - bottom left of p.3, the notation N_0, N_1 , etc is introduced, but does not seem to be used anywhere?

Authors: You are right. The notation was only used to describe the strategy in an explicit way. We agree that this is unnecessary and we modified the main text accordingly.

Referee: - In the comparison between classical and quantum strategies, the authors compare classical variables with an alphabet of size d to quantum systems of Hilbert space dimension d . Why is this comparison appropriate? This should certainly be justified somewhere!

Authors: The main point is that, according to our current understanding of physics, most classical systems (perhaps all) are decohered quantum systems. When we observe a classical process in nature, in general we are observing the

diagonal elements of a quantum process. In this correspondence, d -dimensional classical systems are associated with d -dimensional quantum systems. From this point of view, our results show that preserving coherence gives more powerful ways to detect the causal structure of an unknown physical process.

In the revised version we emphasize the relation between d -dimensional quantum systems and d -dimensional classical systems. To make the point more explicit, we also added a new result on how a coherent superposition of classical states offers an advantage over the best classical strategy.

Referee: Note that if one is comparing the classical alphabet size to the dimension of the quantum state space (d^2) instead of the Hilbert space dimension (d), the quantum advantage would disappear.

Authors: Sure, but so would disappear also the quantum advantages dense coding and entanglement-assisted classical communication. Nevertheless, it is accepted that these results are an important part of our understanding of quantum theory.

Another reason for comparing d -dimensional quantum systems with d -dimensional classical systems is that all correlations mediated by a d -dimensional quantum system can be exactly reproduced with the mediation of a d -dimensional classical system, cf. Phys. Rev. Lett. 119, 020401 (2017). In that work, the authors argue for a task-independent notion of “dimension of physical systems” in which d -dimensional quantum systems are associated with d -dimensional classical systems.

Referee: - top left of p.4, when referring to Figure 3: Fig. 3 does in fact not illustrate the 3 types of strategies, but only parallel ones.

Authors: True. In the revised version this part of the presentation has been reorganized, and now the figure is used to discuss parallel strategies only.

Referee: After Eq. (7), some intuitive explanation is given for the origin of the quantum advantage in terms of the “complementarity between the information about the causal structure and the information about the functional dependence between cause and effect”.

I was wondering if this was just an intuition, or can it be put on firm grounds, e.g. by properly quantifying this complementarity?

Authors: Yes, it can indeed be put on firm grounds. In the revised version we included a complementarity relation between tests of the causal structure and tests of the functional dependence between cause and effect. The relation takes the form of a lower bound on the error probability in identifying the causal structure, expressed in terms of the probability of success in identifying the unitary gate connecting the cause with the effect. The more the probability of success deviates from the random guess probability, the higher is the error probability in the identification of the causal structure.

Thank you for suggesting this addition. We are very pleased with it, because it offers a stronger and more insightful proof of optimality for the parallel strategies without reference. One of the interesting new results following from the complementarity relation is that the error probability of our quantum strategy (working universally for all unitary gates) is optimal even for causal relationships described by arbitrary sets of unitary gates. In fact, it is optimal even if the gate describing the cause-effect relation is perfectly known.

Referee: In Eq. (9): does the notation 'r' refer to the same quantity as the notation 'm_lambda' in the Suppl. Mat.? For clarity one could maybe use only one notation?

Authors: In general, the notations "r" and "m_lambda" denote different quantities: "r" is the rank of the density matrix of the reference system, while "m_lambda" is the dimension of the multiplicity space. For the optimal input state, one has $r = m_\lambda$. In the revised version we clarified these points.

Referee: - Right after Eq. (9): shouldn't one refer to Suppl. Note 4, rather than 3?

Authors: Correct, thank you for spotting the typo. The actual numbering has changed in the revised version, but now the reference after Equation (9) points to the correct Supplementary Note.

Referee: - When referring to the sequential scheme, the authors talk about "the ultimate limit" set by quantum mechanics. Nevertheless, they mention in their conclusion the intriguing question, whether one could further decrease the error probability by using processes with indefinite causal order. If that was the case (with an implementable process, like the "quantum switch"), couldn't be the "ultimate limit" beaten by quantum mechanics? I would then suggest to avoid claiming that they already obtain "the ultimate limit" (or to talk about "the most general test" in the SM).

Authors: Thanks a lot for the wise suggestion. We agree that, if indefinite causal order allowed one to beat the quantum strategy in our paper, then the expression "ultimate quantum limit" would be inaccurate.

Rather than changing the wording, we decided to try and prove that our limit is indeed the ultimate limit allowed by quantum mechanics, no matter if the order is definite or indefinite. We succeeded in this task, proving an explicit lower bound on the error probability of *arbitrary* quantum strategies, even indefinite causal order. No matter what strategy is used, the probability of error cannot go below $1/(4 d^{2N})$, implying that the rate $2 \log d$ is indeed the ultimate quantum limit.

In order to prove the bound, we briefly outlined a general framework for tests with indefinite causal order and used it to derive a semidefinite programming bound on the success probability. Our bound can be regarded as an indefinite-

causal-order analogue of the classical Yuen-Kennedy-Lax bound on state discrimination.

Together with the complementarity relation, the results on indefinite causal order are among the most important additions included in the revised version. We would like to thank you for stimulating our work in these direction.

Referee: In the second part of the Methods, the authors set $U = V = I$, as this is enough to obtain an upper bound to the discrimination rate in the worst case scenario.

However, their goal there is to calculate the fidelity divergence for 2 channels (Proposition 2). Is it then also justified for this calculation to set $U = V = I$? Why?

Authors: Proposition 2 refers to two specific channels with $U=V$. For those channels, the unitary invariance of the fidelity implies that one can choose $U=V=I$ without loss of generality.

Referee: *** In the Suppl. Mat.: Eq. (4), shouldn't it be 'max'?

- 1st line of p.2, the number of the section the authors refer to is missing.

- 2nd line of p.2: should 'Hence' be understood as 'It turns out that...'?

Authors: Agree with all the above points; corrections made.

Referee: - No proof is given for Lemma 2; is that a similar simple adaptation of Holevo's argument?

Authors: That part of the Supplemental Material has been streamlined in the revised version and the old Lemma 2 is not present anymore. But for the records, yes, the old Lemma 2 was a simple adaptation of Holevo's argument.

Referee: - - 3rd line of Suppl. Note 2: 'n' should be a capital 'N'.

- In Lemmas 3 and 4, it may be useful to recall that one assumes N is a multiple of d .

Authors: Correct. This part has changed in the revised version, as the complementarity relation gives us an alternative (and in our opinion more elegant) optimality proof.

Referee: - Eq. (12): does one really need to consider T_{out}^N in the argument? To reach the conclusion of Eq. (14), doesn't one also need some convexity argument?

Authors: You are right that T_{out}^N is unnecessary for our argument. It was mentioned only because it was a symmetry of the channel.

Regarding Eq. (14), the optimality of pure states follows from the fact that the average success probability is a linear function of the state (or that the worst case success probability is a convex function).

Referee: - In Eq. (46) (and then in Lemma 8) I found $c(N) = \prod_{i=1}^d (d-i)!$, i.e., a fixed value independent of N , that does not tend to 1 as N grows to infinity (but remains a constant).

This should be checked again.

If I am wrong and c indeed depends on N , one shouldn't talk about "a constant tending to 1".

Authors: You are absolutely correct. The right statement is " $c(N)$ is a function tending to a constant in the large N limit". We reworded the sentence in the supplemental material accordingly.

Referee: - In Lemma 6, by "Let A be the cause of exactly one variable", do the authors mean it has only one causal intermediary? This may be clarified.

Authors: Yes; clarification added.

Referee: Beginning of the proof of Lemma 6: why is there no loss of generality in considering strategies with all input variables initialised to the same value? This should be justified.

Authors: The argument is the same as in the case of two variables. In the revised version, we explicitly included the proof that a single value is optimal.

Referee: Before Eqs. (61) and (63), I guess it should be " $\lambda_{j/l} \neq \lambda_0$ ", rather than " $\neq 0$ "?

Authors: Correct; thank you for spotting the typo.

Response to Reviewer #2

Referee: In this paper, the authors consider the task of distinguishing different causal relationships, given finite data. They demonstrate that quantum physics allows an exponential reduction of the probability of error, relative to classical physics, for a given number of trials. Specifically, they consider two possible causal relationships among three variables A, B, and C: Either A is a cause of B but not of C, or A is a cause of C but not of B. The variable for which A is not a cause is assumed to be uniformly random.

While the specific problem considered might seem slightly artificial (and the authors do not provide any example of an application - is there one?), the novel demonstration of an exponential quantum advantage in causal inference is very interesting and certainly deserves high visibility and broad dissemination. Causal inference is essential in many branches of science, as pointed out by the authors, as well as a field of research in itself. And the study of causality in the quantum regime is currently highly active. This work is therefore likely to have significant and broad impact, and to spark further work on quantum causal inference.

The manuscript is well written and the analysis is thorough, although with most of the mathematical details relegated to the supplementary material. The derivations are sound, as far as I can tell, although I have not gone through all of them in detail (in particular, I am not familiar with the representations in terms of Young diagrams employed in the supplementary note 3 and onwards), and the techniques quite novel in the context of causal inference, I believe.

I recommend publication in Nature Communications.

Authors: Thank you for the careful assessment and for your positive recommendation towards publication in Nature Communications. We are grateful that you appreciated, in addition to the impact and timeliness of the topic, also the technical innovation of this work relative to previous works in the area of causal inference.

Regarding your question, we believe that the task of distinguishing “A causes B” from “A causes C” has a special interest as one of the simplest instances of two alternative hypotheses about the causal structure. An important feature of this task is that it admits a complete analytical treatment, allowing us to rigorously prove the exponential advantage of quantum strategies over classical strategies.

Thinking of applications, one could think of a network scenario where two nodes, B and C, claim to have a quantum communication link with node A, and the problem is to determine which of the two is the legitimate receiver.

Referee: I do have a couple of questions, which I would ask the authors to clarify, and a few minor comments:

1) In the problem considered by the authors, it is assumed that the causal intermediary (B or C) of A is related to A by an invertible process - i.e. B or C

retains complete information about A. However, it is easy to imagine that this might not be the case in a realistic situation. Can the authors say anything about what happens when the causal intermediary contains only partial information about A? Clearly, in the limit where no information is retained, quantum and classical strategies must perform equally bad. Is there still a quantum advantage in the regime in between?

Authors: We agree with you that the noisy scenario is important, although we also would like to stress that the emphasis of our paper is on the in-principle advantage reachable in the noiseless scenario. The situation is similar to that of quantum metrology, where the first works highlighted the possibility of achieving a non-classical scaling in the noiseless scenario, and later works addressed more realistic situations involving noise.

In the revised version, we discuss some noisy scenarios where the quantum advantage still persists. One easy example is the correlated noise where a random unitary gate acts identically on all input systems. In this case, probability of error of our quantum strategy is the same as in the noiseless scenario, and therefore the same quantum advantage applies. A less trivial example involves cause-effect relationships described by the depolarizing channel, which leaves the state unchanged with probability $(1-p)$ and replaces it with a maximally mixed state with probability p . For the causal hypotheses considered in our paper, this is a “bad” type of noise, because it probabilistically makes the causal hypotheses indistinguishable. Despite this fact, we show that the quantum advantage survives even in the presence of depolarization.

Finally, we also provide a partial analysis of the more challenging setting where the cause-effect relation is described by a depolarizing channel combined by a completely unknown unitary gate. While the full analysis is beyond the scope of this paper, we provide a proof of quantum advantage in the simplified scenario where the depolarization is heralded. While this example is not directly of practical interest, it does prove that some types of uncorrelated noise can be benign to the detection of causal relations.

Referee: 2) Below Eq. (9), the authors explain that the number of linearly independent states entering in the sum in Eq. (8) grows d^N . I think this claim is based on the argument around Eqs. (45)-(47) in the supplementary material. However, it is not obvious to me why the statement about $m_{\{\lambda_0\}}$ proven there implies the claim in the main text. Could the authors explain this?

Authors: In the revised version we make this point explicit, proving that the state written down in the main text coincides with the optimal state obtained in the group-theoretic framework, and that the number of linearly independent states is indeed equal to $m_{\{\lambda_0\}}$. The details are now presented in Proposition 8 of Supplementary Note 2.

Referee: 3) In the review of previous work in the introduction and the discussion of interventions, it would appear relevant to reference work showing that the quantum behaviour under interventions can be very different than the

classical one, for fixed scenarios. E.g.

- Chaves, Carvacho, Agresti, Di Giulio, Aolita, Giacomini, Sciarrino, Nature Physics 14, 291 (2018).

- Van Himbeeck, Brask, Pironio, Ramanathan, Sainz, Wolfe, arXiv:1804.04119v1 [quant-ph] (2018).

Authors: Thank you for the very appropriate references; we have included both in the revised version.

Referee: 4) On p. 4 the authors write that "The three type of strategies corresponding to the above three features are illustrated in Figure 3". However, the third strategy (coherence in time) doesn't seem to be illustrated in Fig. 3?

Authors: In the revised version we modified this part.

Referee: 5) Below Eq. (9), I think N in the exponent should be N/d ?

Authors: Thank you for spotting the typo.

Referee: 6) The formulation in the outlook that quantum physics may be "at least, never worse" than classical physics is perhaps a bit confusing, as, for many purposes, classical physics is contained in quantum physics as a special case. In that case, obviously quantum can be no worse than classical. I guess the authors maybe refer to the challenges when genuinely quantum causal structures need to be identified.

Authors: Yes, this is indeed what we meant. When doing the revisions we found convenient to omit that comment, because the paper contains many new results and we decided to carve out more space to discuss them.

Referee: 7) In Lemma 1 on p.2 of the supplementary material, I find the notation a bit strange. The authors refer to the worst-case error probability for distinguishing channels $C_{\{1,V\}}$ and $C_{\{2,U\}}$. But the worst-case error probability refers to the minimisation over U and V (as in Eq. (4)), and so does not depend on U or V . Neither does the average error probability. So perhaps in the lemma it would be better to leave U and V out of the subscripts.

Authors: We agree. In the revised version, Supplementary Notes 1 and 2 have been completely rewritten in the light of new results. In the process of rewriting, we made an effort to make the presentation simpler and more accessible.

Referee: 8) Below Eq. (38) in the supplementary material, perhaps it would be clearer to say that $m_{\{\lambda_0\}}$ has to be exponentially large rather than "asymptotically" large.

Authors: Absolutely; thanks for the suggestion.

Referee: 9) In the Methods section, the authors explain that the third line in Eq. (16) follows from the definition of the fidelity divergence. However, this was not

obvious to me. Could the authors elaborate?

Authors: In the revised version we added a comment (new Eq. (16)) to clarify this point.

Response to Reviewer #3

Referee: First of all I would like to write that I have read submitted manuscript 'Quantum speedup in testing casual hypotheses' by Giulio Chiribella and Daniel Ebler with the great interest.

Authors: Thank you for your interest, we are glad that the manuscript has been an engaging reading.

Referee: Authors in their manuscript prove that quantum physics offers an exponential advantage over classical one in the task of testing casual hypotheses. Namely, the main result states that quantum error probability of guessing incorrect hypothesis is d times smaller than in the classical counterpart.

Authors: There seems to be some confusion here. The main result is not that the quantum error probability is d times smaller than its classical counterpart. The main result is that the quantum error probability is d^N times smaller than its classical counterpart (up to a polynomial factor). This advantage is exponential in the number N of times the unknown process is sampled, and implies that quantum causal relationships can be identified with roughly half of the queries needed by classical strategies.

Referee: In all proofs Authors extensively use tools coming from representation theory of $SU(d)$ and $S(n)$, in particular famous Schur-Weyl duality. Symmetries occurring in the problem allowed to find analytical expressions for optimal states and error probabilities in four different scenarios. I have to emphasize here that whenever we ask about optimal quantities in some protocols, obtaining answer in closed form, in other words being able to rid of optimisation over all parameters etc. is a very hard task.

Authors: Thank you for appreciating the difficulty of the problems addressed in our paper and the work that went into solving them.

Referee: In my opinion all proofs in appendices are correct. They are written clearly and explicitly, allowing potential Reader to follow all key steps in the argumentation.

Authors: Thank you for confirming that the appendices are helpful to the Reader. In the revised version we further streamlined them, hopefully making the results even more accessible.

Referee: Additionally, I have a few minor remarks (according <https://arxiv.org/abs/1806.06459>):

1) In my opinion there is no deep motivation for considering such kind of problem(s) through whole introductory section. Of course, I am aware of the main goal is better understanding of surrounding World and differences between quantum and classical regimes. But for such high-impact journal we

would expect much more deeper motivation and implications, rather than some general remarks and conjectures.

Authors: We understand your point of view. Still, it is important to stress that the above comment looks more like a criticism to the whole research area of quantum causality, which is indeed motivated by a general interest in understanding causal relations in the quantum domain.

The motivation of our paper is the same as that of the papers we cite in the introduction, including papers that have indeed been published in Nature Communication, Nature Physics, or other high-impact journals. The presence of this line of research in high-impact venues suggests that the general questions asked in this area are of interest to a broad community. It is still true, as you say, that the concrete applications of quantum causal discovery algorithms are still to be worked out, and the implications of quantum causality to other areas of quantum physics are mostly unexplored. Yet, this seems to be normal, given that the whole line of research is fairly new.

More specifically to our work: this work fulfills a promise that was implicitly made in Nature Physics 11, 414 (2015), namely that quantum strategies can be more efficient at detecting causal relations. Our results answer a question that many colleagues in the community had upon reading that paper: is there a quantum advantage when we allow for the most general classical strategies for detecting causal dependencies? This question is rather fundamental, and the techniques we develop to address it are likely to be useful in other problems of quantum causal inference. In the revised version we provide some evidence in this direction, showing that our techniques can be used to obtain quantum advantages in tasks other than the identification of the effect of a given variable.

Having said this, we took your comment seriously, and made an effort to articulate more explicitly why the questions addressed in this paper are important. The motivational part is complemented by new results, which illustrate the applicability of our framework.

Referee: What exactly factor $1/d$ offers to us? It is well known that in many situations quantumness offers more than classicality, and I would like to know why, in particular this result deserves for publication in Nature Communication.

Authors: Once again, let us stress that the advantage is *not* of a factor $1/d$, but of a factor $1/d^N$. The crucial point is that the advantage is *exponential* in N , the number of experiments. This means that quantum physics offers us an exponentially faster way to detect causal relations compared to classical physics. This advantage applies both to the fundamental problem of detecting causal relations and to the more practical problem of exploring the connectivity of a quantum communication network: we can imagine that in a future quantum internet a local user will need to test his/her connection, trying to “ping” other nodes of the network and to see if there is a connection or not. Among quantum technologies, quantum communication networks are probably one of the most

mature, and having a fast algorithm to detect which node communicates with which is likely to be a useful primitive in a future quantum internet.

Referee:

- 2) Full stop is missing in the first paragraph on the page 4. Just before the sentence 'The three type of strategies...'
- 3) In the fourth line of equation (16) it should be ∂ instead of δ .
- 4) It should be ϕ_{xn} instead of ψ_{xn} in the text, between equations (19) and (20).
- 5) In the sentence below equation (57) the reference is needed.
- 6) There is missing comma just after expression in equation (59).
- 7) Full stop just after expression in equation (113) is unnecessary.

Authors: Thank you for spotting the above typos and for suggesting improved wordings. We incorporated all your suggestions in the revised version.

Referee: 8) For me considerations in Appendix G look similar to probabilistic port-based teleportation (in the context of signals discrimination). But somehow probability of success behaves differently. Could Authors elaborate on this?

Authors: Thanks a lot for this question. The connection was not immediate, and your question stimulated us to extend our work to a new causal hypothesis scenario.

As it turns out, the connection is with *deterministic*, rather than *probabilistic* PBT. It is well known that the teleportation fidelity of deterministic PBT is equal to the probability of successfully distinguishing a certain set of states. In the standard PBT protocol, the states are obtained from k maximally entangled states by tracing out one of the two entangled systems in all but one of the pairs. This problem is equivalent to a causal discrimination task where there are k input variables and one output variable, with the promise that one and only one of the input variables is a cause for the output variable.

This problem is trivial when the number of hypothesis k is smaller than the dimension d of the variables. The interesting regime is when the dimension is fixed and the number of hypotheses is large. In this regime, the best classical strategy needs to sample the unknown process for $\log_d k$ times. Instead, the connection with deterministic PBT allows us to prove that a quantum strategy can identify the unknown cause with approximately half of the queries, and vanishing error probability in the large k limit. The proof uses a recent result by Mozrzykas, Studzinski, Strelchuk, and Horodecki [New Journal of Physics 20, 053006 (2018)], which is adapted here to the setting where the cause-effect relationship is a completely unknown gate.

We gratefully acknowledge your question, which led us to this new result. Identifying the cause of a given effect is a very fundamental task, and having a provable quantum speedup in this task provides further support to the point made by our paper.

Referee: Results presented in the manuscript are interesting and correct, whole paper is a nice and very solid piece of work, but in this form probably it will not be interesting for broader audience, even in the field of quantum information theory. I strongly recommend to include more detailed discussion on implications of obtained results or find other, specialised journal for the publication. Having additional results I would be happy to reconsider manuscript for the publication in Nature Communication.

Authors: Thank you for the appreciation of the technical strength of the manuscript and for your openness to reconsider your assessment on the impact. Needless to say, we are excited about the science presented in the manuscript, and we hope to convey to you the reasons why we believe it is important to give high visibility to these findings.

The revised version does indeed contain new results, the most notable of which are:

(1) A new complementarity relation between tests of causal structure and tests of the functional dependence between cause and effect. This result provides an intuitive explanation for why the optimal tests of causal structure should be “blind”, in the sense that they should only extract information on which variable depends on which, without trying to discover the specific way in which the effect depends on the cause.

(2) A proof that the asymptotic performance of our quantum strategy is (within a polynomial factor) the ultimate limit allowed by quantum mechanics, even if one allows exotic operations where the unknown process is tested in an indefinite order.

To the best of our knowledge, this is the first time that a non-trivial bound on the distinguishability power of indefinite causal order has been proven. In order to obtain this result, we built the appropriate theoretical framework of “*indefinite testers*”. Our framework provides the foundation for a new research direction on quantum metrology with the assistance of indefinite causal order.

In addition to the framework, we derived a general semidefinite programming bound on the success probability of strategies with indefinite order, generalizing the classic bound of Yuen, Kennedy, and Lax for state discrimination.

(3) Quantum advantages in new scenarios. We show that the techniques developed in the main example of the paper can be generalized to other tasks of causal hypothesis testing. This includes:

(3.1) the task of distinguishing whether there exists a causal relationship between two variables. Having a fast quantum strategy for this task could be useful for application in quantum communication, e.g. as a way to explore the connectivity of an untrusted communication network.

(3.2) the task of deciding which variable is the cause of a given effect. Arguably, this is a rather fundamental problem, and having a quantum speedup in it could have applications in some tasks of quantum machine learning, in the same way as causal discovery algorithms have applications in classical Bayesian learning.

This addition was stimulated by your question about the connection of our work with port-based teleportation, and we are pleased to thank you for having stimulated our work on this front.

REVIEWERS' COMMENTS:

Reviewer #1 (Remarks to the Author):

The authors have satisfyingly addressed all the points I raised in my report, and, as far as I can tell, the points raised by the other referees. They have added quite interesting results to their earlier version. I am now happy to recommend the paper for publication in Nature Communications.

Reviewer #2 (Remarks to the Author):

In their revised manuscript and accompanying reply, I find that the authors have addressed all my previous concerns in a satisfactory manner (though, in the reply, I guess by "Proposition 8 of Supplementary Note 2" they meant Proposition 4). In particular, they now provide some analysis of the case where the causal intermediaries do not retain full information (a noisy scenario, as the authors call it), with new proofs added to the supplementary material.

I therefore do not hesitate to recommend publication.

I also note that the other Referees raised several interesting points, to which the authors, in my view, provide thorough and convincing answers and additions to the manuscript.

Jonatan Bohr Brask

Reviewer #3 (Remarks to the Author):

Dear Editors,
Dear Authors,

First of all I am really grateful Authors for clarifications, taking all my comments and suggestion into consideration. Now, in my opinion the importance of the area of research chosen by the Authors is better motivated for the common reader. Also, I appreciate the new results derived using recent developments in port-based teleportation.

Summarising, having additional results and improvements I am happy to recommend manuscript for the publication in Nature Communications. Many congratulations!

Response to Reviewer #1

Referee: The authors have satisfyingly addressed all the points I raised in my report, and, as far as I can tell, the points raised by the other referees. They have added quite interesting results to their earlier version. I am now happy to recommend the paper for publication in Nature Communications.

Authors: Thank you for your positive recommendation, and again, for the stimulating suggestions that helped improving the original manuscript.

Response to Reviewer #2

Referee: In their revised manuscript and accompanying reply, I find that the authors have addressed all my previous concerns in a satisfactory manner (though, in the reply, I guess by "Proposition 8 of Supplementary Note 2" they meant Proposition 4).

Authors: We did mean Proposition 4, indeed. Apologies about the confusion, due to the restructuring of the Supplemental Material, whereby some propositions from Supplementary Note 8 were removed.

Referee: In particular, they now provide some analysis of the case where the causal intermediaries do not retain full information (a noisy scenario, as the authors call it), with new proofs added to the supplementary material.

I therefore do not hesitate to recommend publication.

Authors: Thank you for your positive recommendation, and for stimulating our work on the "noisy scenario". We believe there is still a substantial amount of interesting research to be done in that direction.

Referee: I also note that the other Referees raised several interesting points, to which the authors, in my view, provide thorough and convincing answers and additions to the manuscript.

Jonathan Bohr Brask

Authors: Thank you for looking into the other Referees' suggestions, and into our answers. We were indeed pleased to have this opportunity to further enrich our manuscript, and are thankful to all Referees for their stimulating comments.

Last but not least, we would like to thank you for choosing to provide an open review, we greatly appreciate it.

Response to Reviewer #3

Referee: First of all I am really grateful Authors for clarifications, taking all my comments and suggestion into consideration. Now, in my opinion the importance of the area of research chosen by the Authors is better motivated for the common reader. Also, I appreciate the new results derived using recent developments in port-based teleportation.

Summarising, having additional results and improvements I am happy to recommend manuscript for the publication in Nature Communications. Many congratulations!

Authors: Thank you for your open mindedness in considering and embracing our additions to the original manuscript. We are delighted that the revised version of our manuscript met your requests.

Thank you also for prompting us to look into port-based teleportation, this has been an important opportunity to strengthen the general message of our work by providing further examples of quantum advantages in the identification of causal relationships. The connection between port based teleportation and strategies for identifying causal structures is also likely to be a fruitful topic of future research, and we are grateful for your suggestion, which stimulated us to consider this direction.